# Arabic Mini-ClimateGPT :
# A Climate Change and Sustainability Tailored Arabic LLM

Sahal Shaji Mullappilly[1]∗    Abdelrahman Shaker[1]∗    Omkar Thawkar[1]∗    Hisham Cholakkal[1]
Rao Muhammad Anwer[1]    Salman Khan[1]    Fahad Shahbaz Khan[1,2]

[1]Mohamed bin Zayed University of Artificial Intelligence    [2]Linköping University

## Abstract

Climate change is one of the most significant challenges we face together as a society. Creating awareness and educating policy makers the wide-ranging impact of climate change is an essential step towards a sustainable future. Recently, Large Language Models (LLMs) like ChatGPT and Bard have shown impressive conversational abilities and excel in a wide variety of NLP tasks. While these models are close-source, recently alternative open-source LLMs such as Stanford Alpaca and Vicuna have shown promising results. However, these open-source models are not specifically tailored for climate related domain specific information and also struggle to generate meaningful responses in other languages such as, Arabic. To this end, we propose a light-weight Arabic Mini-ClimateGPT that is built on an open-source LLM and is specifically fine-tuned on a conversational-style instruction tuning curated Arabic dataset *Clima500-Instruct* with over 500k instructions about climate change and sustainability. Further, our model also utilizes a vector embedding based retrieval mechanism during inference. We validate our proposed model through quantitative and qualitative evaluations on climate-related queries. Our model surpasses the baseline LLM in 88.3% of cases during ChatGPT-based evaluation. Furthermore, our human expert evaluation reveals an 81.6% preference for our model's responses over multiple popular open-source models. Our open-source demos, models and curated instruction sets are available here https://github.com/mbzuai-oryx/ClimateGPT.

## 1    Introduction

Climate change poses a significant and urgent challenge in our modern era, as it has far-reaching and profound consequences for the environment, ecosystems, economies, and human well-being.

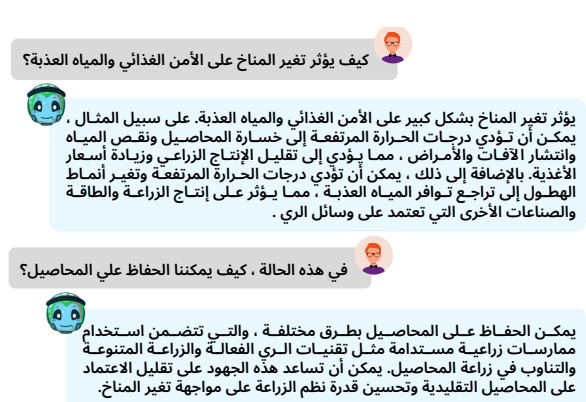

Figure 1: **Arabic Mini-ClimateGPT conversation**: **User**: *How does climate change affect food security and freshwater?* **Response**: *Climate change has a significant impact on food security and freshwater. For example, high temperatures can lead to crop loss, water scarcity, the spread of pests and diseases, resulting in reduced agricultural production and increased food prices. Additionally, high temperatures and changes in precipitation patterns can lead to a decline in the availability of freshwater, affecting agricultural production, energy, and other industries that rely on irrigation methods.***User**:*In this case, how can we preserve the crops?* **Response**:*Crops can be preserved through various methods, which include using sustainable agricultural practices such as efficient irrigation techniques, crop diversification, and crop rotation. These efforts can help reduce reliance on traditional crops and enhance the resilience of agricultural systems to climate change.* (**best viewed in zoom**)

Educating people about climate change and sustainability is essential for preserving the environment, encouraging sustainable development, and preparing for future challenges. For improving public awareness on climate change and sustainability related issues, conversational agents and bots can play a vital role. Despite being spoken by around 422 million (Wikipedia contributors, 2023) speakers worldwide, efforts towards developing climate-specialized conversational agents in Arabic have been limited. In this work, we propose a specialized conversational agent in Arabic focused on climate change and sustainability awareness. Such a specialized conversational agent can have a wide range of applications. For instance, it can be utilized to educate students about climate change by generating relevant educational materials. Furthermore, environment policy makers can leverage such an agent to make well-informed and sustainable decisions.

Recently, large language models (LLMs) such

---

∗Equal Contribution

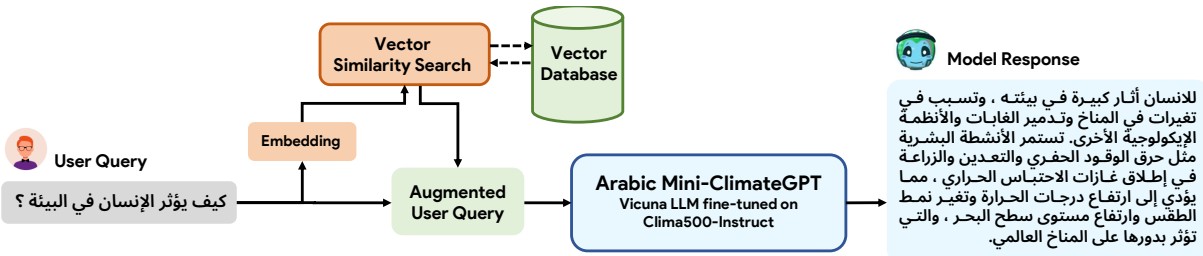

Figure 2: **Overview of Arabic Mini-ClimateGPT framework**: When the user submits a query, it is first embedded and searched for similarity in the vector database. The query is then augmented if the retrieved information is of high correspondence and passed on to our model for generating the final output.

as ChatGPT(OpenAI, 2022) from OpenAI and Bard(Google, 2023)from Google have demonstrated excellent capabilities to behave like versatile conversational agents and generate responses for a wide range of tasks. However, it is important to note that Bard and ChatGPT models are currently not available as open-source, thereby limiting access to their underlying architecture and implementation details. The recently introduced open-source model Vicuna (Chiang et al., 2023) demonstrates exceptional performance and achieves more than 90% of ChatGPT's quality on GPT-4 evaluations. The model was created by fine-tuning LLaMA (Touvron et al., 2023) and surpasses existing open-source competitors across various benchmark evaluations. We base our climate-specialized conversational agent in Arabic, named Arabic Mini-ClimateGPT on Vicuna.

The main contributions of this work are :
**(i)** We propose a climate specialized Arabic conversational agent named Arabic Mini-ClimateGPT, which is built upon Vicuna framework and fine-tuned specifically with our climate change and sustainability related instructions in Arabic language.
**(ii)** We have generated over 500k conversational-style instruction tuning data based on the public benchmarks for climate change and sustainability. This augmentation of interactive conversational data significantly enhances the performance of LLMs and preserves its generalizability through the fine-tuning process. To the best of our knowledge, our proposed dataset *Clima500-Instruct* marks the first release of a substantial conversational-style Arabic instruction set dedicated to Climate change and sustainability.
**(iii)** We integrate a vector embedding and datastore framework, which can be utilized during model inference for information retrieval without the need for additional training.
**(iv)** We perform comprehensive evaluations of our

model. Our model achieves 88.3% win rate in ChatGPT-based comparison with baseline. Furthermore, our Arabic language expert evaluation shows that our model responses were preferred in 81.6% of the cases, whereas popular open-sourced models such as Vicuna, Alpaca-Arabic and Dolly v2 were successful only in less than 8% of the cases.

## 2   Related Work

ClimateQ&A (Ekimetrics, 2023) is a ChatGPT based tool that can distil climate science knowledge based on IPCC reports and environment articles to user friendly responses. Although this model is available on HuggingFace, it is not open-sourced as it is dependent on ChatGPT. Moreover, its responses are limited to English. ClimateBot (Rony et al., 2022) is another machine reading comprehension bot for Question Answering about climate change documents. It is limited to localizing the answer from a given context as it is based on a clustering approach and lacks the generative capabilities of modern LLMs. The model responses and document embeddings are limited to English. To the best of our knowledge, we are the first to introduce an open-sourced, climate-specialized conversational agent in Arabic.

### 2.1   Data Sources

**CCMRC:** Climate Change dataset (Rony et al., 2022) is a climate and sustainability question-answer dataset curated from trusted sources including the official Semantic Scholar Dump, the IPCC Reports, NASA Global Climate Change and individual documents. The dataset also includes news articles from several sources like CNN, National Geographic and The New York Times. The data was pre-processed into 21K Question Answer pairs used for training and testing in our experiments.
**ClimaBench:** This is a benchmark dataset (Laud et al., 2023) for Climate change text understand-

ing, which is built upon the publicly available surveys released by CDP (formerly the Carbon Disclosure Project (CDPInc, 2000)) and the NAIC Climate Risk Disclosure Survey. ClimaBench consists of Clima-Insurance and Clima-CDP for text classification and Clima-QA for question answering tasks based on the CDP data. We use this Clima-QA data compiled from the CDP-Cities, CDP-States and CDP-Corporation subsets and preprocess into 485K Question Answer (QA) pairs. The proposed Arabic Clima500-Instruct dataset is built upon CCMRC and ClimaBench datasets.

## 3 Arabic Mini-ClimateGPT

Our Arabic Mini-ClimateGPT is an open-sourced conversational agent instruction-tuned for Climate change and Sustainability in Arabic. We first create a conversational style instruction-tuning dataset named *Clima500-Instruct* in Arabic language based on CCMRC and ClimaBench (See Sec. 3.1). Our Arabic Mini-ClimateGPT is built upon the Vicuna-7B model by fine-tuning on the *Clima500-Instruct*. We also integrate a dedicated Inference module to incorporate region-specific documents in Arabic to extend the knowledge base of our model. (See Sec. 3.2)

### 3.1 Arabic Clima500-Instruct Dataset

We propose a conversational style instruction-tuning dataset named *Clima500-Instruct* for Climate change and Sustainability in Arabic language. **Dataset Creation:** The existing climate-specific datasets such as CCMRC and Clima-QA from ClimaBench are naive QA datasets in English that lack the interactive conversation capabilities required to train a conversational agent such as Vicuna. To this end, our proposed Clima500-Instruct dataset is a conversational-style instruction tuning dataset in Arabic language with Climate change and sustainability as the central theme. Fig. 3 shows our overall dataset creation pipeline which is summarized below:
**(i)** We first pre-process the data and extract the Question Answer (QA) pairs from their respective sources. These QA pairs are then compiled and converted to a naive QA instruction set.
**(ii)** We then instruct ChatGPT to act as a domain expert in Climate change and sustainability to *generate conversational-style responses in English* grounded in information based on the QA pairs from CCMRC and ClimaBench.

| Dataset | Total Instances | Avg. Question length (token) | Avg. Answer length (token) |
|---|---|---|---|
| Clima500-Instruct (Arabic) | 506426 | 49 | 939 |
| Clima500-Instruct (English) | 512081 | 23 | 196 |

Table 1: **Clima500-Instruct Dataset Statistics**

**(iii)** The next task involves *translating this instruction set into Arabic*. Rather than resorting to rudimentary translation APIs, we employ the capabilities of ChatGPT. We instruct ChatGPT to operate as an expert in the Arabic domain, translating both the question and the answer within a single prompt, thereby facilitating improved contextual understanding by the model.
**(iv)** Finally, we *manually filtered out low-quality examples* and post-processed the instruction set with the assistance of native Arabic experts. The following steps were taken to post-process the dataset. (a) Eliminate undefined symbols. (b) Verify and manually translate English words in several questions and answers into Arabic. (c) Remove any instances of corrupted translation (Q or A still being in English). (d) Verify the translation for N random samples. This approach enabled us to obtain a high-quality, conversationally-styled instruction tuning dataset, *Clima500-Instruct*, in the Arabic language.
**Dataset Statistics:** Table 1 summarizes the statistics of our proposed Clima500-Instruct dataset in Arabic language as well as its intermediate byproduct in the English language. The train and test set splits of our Clima500-Instruct dataset are carefully prepared based on the respective splits of representative data sources mentioned in section 2.1. To further understand the composition of our proposed Clima500-Instruct Dataset, we obtain high-level categories in our dataset based on the most frequent words. Table 2 reports the categories and their corresponding instance counts.

### 3.2 Model Training and Inference

Our Arabic Mini-ClimateGPT is developed on top of the Vicuna-7B (Chiang et al., 2023) model. We fine-tune this Vicuna-7B model on the conversational style Clima500-Instruct in both English and Arabic languages. We train the model on 4 x A100 (80GB) utilizing gradient checkpointing and flash attention following the Vicuna fine-tuning recipe. We observe that the Arabic ClimateGPT model requires a context length of 1024 tokens to generate high quality results. The overall framework of our model with its inference time retrieval capabilities is shown in Fig. 2.

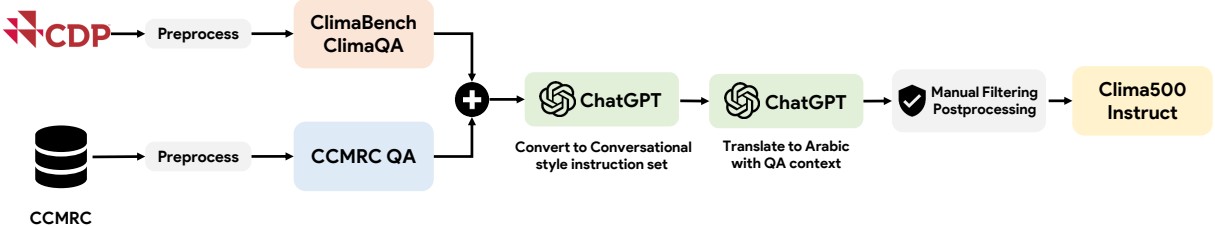

Figure 3: **Clima500-Instruct Arabic Dataset creation pipeline**: At first, CCMRC and ClimaBench dataset are pre-processed and compiled into QA pairs. These QA pairs are then converted to a conversational-style instruction set and further translated into Arabic using ChatGPT. The resulting instruction set undergoes manual verification and post-processing to yield the final *Clima500-Instruct* dataset. (Refer 3.1)

| Categories | Instance Count |
|---|---|
| Temperature | 28796 |
| Precipitation | 24603 |
| Oceanic | 75303 |
| Extreme weather | 45506 |
| Land cover | 14845 |
| Greenhouse Emissions | 290925 |
| Hydropower / Hydrology | 16169 |
| Air Quality / Index | 10122 |
| Renewable Energy | 6013 |
| Climate Policy / Laws | 391403 |
| Other | 124678 |

Table 2: **Clima500-Instruct high-level categories and their instance count**

**Inference Module:** Large Language models have shown remarkable capabilities for a variety of NLP tasks in recent times. However, their knowledge-base is inherently limited to their training data. Recent frameworks (Chase, 2022) have shown that the use of Vector embedding based retrieval mechanisms can be leveraged to extend the knowledge base of an LLM. To this end, we integrate a vector database using the ChromaDB(Huber et al., 2020) framework. We use the *stsb-xlm-r-multilingual* model from Sentence Transformers (Reimers and Gurevych, 2019) to embed the Arabic Climate documents to leverage its multilingual embedding capabilities. Climate reference documents in English are converted to vector embedding using the lightweight *all-MiniLM-L6-v2* from Sentence Transformers (Reimers and Gurevych, 2019).

The inference steps of our model can be summarized as the following. When a user submits a query to the model it is first converted into the embedding space of the corresponding Sentence Transformer (Reimers and Gurevych, 2019) based on the language. We then perform a Vector similarity search of this query embedding with the existing Vector Database using ChromaDB. If the similarity search returns a high correspondence, the user query is then *augmented* with the retrieved additional context. If the retrieved documents are not relevant then the original query is passed to the Arabic Mini-ClimateGPT model. Our model fine-tuned on the Arabic Clima500-Instruct dataset, then generates the final output by incorporating the supplementary retrieved context (See Fig 2).

## 4 Experiments

As LLMs become more sophisticated, the existing open benchmarks may no longer suffice to adequately assess their capabilities. These benchmarks are typically designed to test chatbots on a limited set of tasks, such as question-answering or context-based localization. To address this, we compare the model responses with ground truth responses on a held-out Clima500-Instruct Arabic test set using ChatGPT. We compare the generated responses of our model with other open-source LLMs like Vicuna(Chiang et al., 2023), Stanford Alpaca (Taori et al., 2023), Dolly v2 (Databricks, 2023) and Arabic Alpaca(Yasbok, 2023) on the test set. For each question within the test set, we provided ChatGPT with the following inputs: (i) the ground-truth answer text, (ii) the response texts generated by our model, and (iii) the text generated by the designated competitor model. In this context, ChatGPT's role was to assess and contrast the text outputs of our model and the competitor model, subsequently selecting the response that best semantically aligns with the ground truth. If neither of the generated texts semantically matches the ground truth, ChatGPT outputs "Neither" as the response. The row headers in Table 3 indicate the

occurrences of "Our," "Competitor," and "Neither" being favored, alongside the corresponding win percentages. For instance, in the first row, our model is preferred in 88.33% of test samples, Vicuna is favored in 11.23%, and in 0.43% of cases, neither our model nor Vicuna is preferred. Our Arabic Mini-ClimateGPT model outperforms the recently introduced Vicuna with a win rate of 88.3% on ChatGPT evaluation. (See Tab 3). We also perform human expert based evaluation on our Arabic Clima500-Instruct test set with native Arab speakers. For Human Evaluation to be in a fair setting, we remove all model identifiers and provide responses from all 5 models to Arab native speakers. They then select the best response out of the 5 (or none) for a given question. The evaluation results on recently introduced open-source models are shown in Fig 4. Our Arabic Mini-ClimateGPT achieves an overall win rate of 81.6% on human expert evaluations.

Table 4 presents the comparison results of naive translated baselines with our proposed model. We employ the same ChatGPT-based evaluation methodology, comparing our proposed Arabic Mini-ClimateGPT responses against these baseline models. (i) *Vicuna_tr*: In this baseline, we begin by translating the query from Arabic to English. Next, we generate the model response in English using the Vicuna baseline. Finally, we translate the responses back to Arabic. (ii) *ClimateGPT_en_tr*: In the second baseline, we also execute Arabic-English and English-Arabic translations. However, instead of the Vicuna baseline, we utilize ClimateGPT_en. This model has been trained using the our proposed Clima500-Instruct in English. In both scenarios, we observe that our Arabic Mini-ClimateGPT responses outperform these baselines. Our model achieves 73.98% win rate compared to Vicuna_tr with only 25.84%. In case of ClimateGPT_en_tr (30.07%), our model achieves 69.57% win rate on ChatGPT evaluations. These results demonstrate the necessity for a dedicated Arabic conversational dataset and model learning.

## 5   Conclusion

In this paper, we present Arabic Mini-ClimateGPT, a climate-specialized conversational agent in Arabic. Our framework incorporates a vector embedding based retrieval mechanism to extend the knowledge-base of our model during inference. Our proposed dataset Clima500-Instruct is an ex-

| Model | Ours | Competitor | Neither |
|---|---|---|---|
| Vicuna | 88.33% | 11.23% | 0.43% |
| Alpaca | 90.57% | 8.81% | 0.61% |
| Dolly v2 | 90.23% | 8.81% | 0.95% |
| Alpaca-ar | 89.71% | 9.42% | 0.86% |

Table 3: **ChatGPT Evaluation**: Our Arabic Mini-ClimateGPT compared with other open-source models, where ChatGPT was tasked to pick the best response based on the ground truth. Our model achieves nearly 90% win rate with all the competing models including the Alpaca-ar in Arabic language.

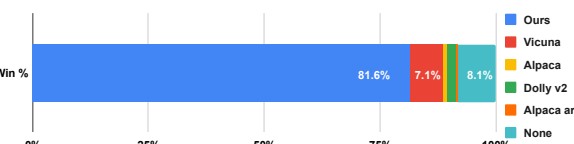

Figure 4: **Human Evaluation**: We perform evaluation on various open-source models on our Clima500-Instruct test set with native Arab speakers. Win % of each model is illustrated in the graph along with their respective legends. Our model achieves *81.6%* better responses compared to other models in Human evaluation.

| Model | Ours | Competitor | Neither |
|---|---|---|---|
| Vicuna_tr | 73.98% | 25.84% | 0.17% |
| ClimateGPT_en_tr | 69.57% | 30.07% | 0.34% |

Table 4: **Comparison with Naive baseline with translated responses.** *Vicuna_tr* stands for baseline Vicuna with translated responses. *ClimateGPT_en_tr* stands for English ClimateGPT with translated responses.

tensive collection of conversational style instruction tuning data dedicated to sustainability and climate change in Arabic language. We open-source our model weights and instruction set and we hope that our work helps advance the community towards a sustainable future.

## 6   Limitations

Our model is developed upon Vicuna, a model based on the LLaMA architecture. The LLaMA model has been trained on a range of data sources, such as CommonCrawl, C4, and Wikipedia. It should be noted that these sources may contain inaccurate information. The model also could be tuned to show racial biases by modifying the prompt, a potential limitation that users should be aware of. Furthermore, our model can sometimes generate unrealistic or 'hallucinated' responses when conversations exceed the maximum

context length. Examples of failure cases are shown in Appendix A.

Currently, our Arabic Mini-ClimateGPT is limited to the language domain and cannot incorporate visual data. Recently introduced Large Language-Vision Models (LLVMs) (Zhu et al., 2023; Liu et al., 2023; Thawkar et al., 2023; OpenAI, 2023; Muhammad Maaz and Khan, 2023) have demonstrated promising results in several domains. We hope that integrating a dedicated Vision encoder for Climate related modalities (e.g. temperature, wind, precipitation) with our model would be a promising future direction.

## 7 Ethics Statement

In alignment with the ACL Ethics Policy, we address the ethical dimensions of our work on Arabic Mini-ClimateGPT. We have conscientiously credited the data sources and other open source works on which Arabic Mini-ClimateGPT is built upon. We have worked diligently to reduce biases in model responses and improve the credibility of generated responses through the implementation of our VectorDB module. Our commitment to transparency is evident through the availability of open-source resources, and we value collaboration and accountability within the research community. In recognizing the broader societal impact of our research, we pledge to uphold ethical standards in the development and deployment of our model for Climate and sustainability related information dissemination.

## 8 Acknowledgement

We would like to thank our colleagues at MBZUAI for their essential contribution to the Evaluation and Dataset verification tasks, including Dr. Jean Lahoud, Abdelrahman Shaker, Salwa Al Khatib, Mohamed El Amine Boudjoghra, Aisha Fahad Ahmed Ali Alraeesi, Amna Abdelrahim Nasir Abdalla Alhosani, Hour Eisa Abdelrahim Ahmed Mohamed, Hosam Mahmoud Abdalla Ahmed Ali Elgendy, Yahia Dalbah, Mohammed Almansoori, without which this project would not be possible.

The computational resources were provided by the National Academic Infrastructure for Supercomputing in Sweden (NAISS), partially funded by the Swedish Research Council through grant agreement no. 2022-06725, and by the Berzelius resource, provided by the Knut and Alice Wallenberg Foundation at the National Supercomputer Centre.

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

# A Appendix

## A.1 Qualitative Examples

Qualitative examples of our Arabic Mini-ClimateGPT compared to the baseline Vicuna are shown from Fig 5 - 7. The responses generated by our model are more coherent and factual compared to Vicuna. Please refer Fig 5 - 7 for more details.

## A.2 Examples of Failure Cases

Since our model is a domain fine-tuned LLM, its capabilities are limited to the Climate change and sustainability domain. In Example A (Fig 8) we can see that the model cannot handle out of domain questions. In Example B (Fig 9) we can see that the model is susceptible to Prompt hacking. In both cases, the model gives a generic response without addressing the query.

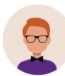

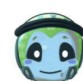

ما هي الآثار المتوقعة لتغير المناخ على المدى القريب والبعيد؟

ستكون الآثار المتوقعة لتغير المناخ على نطاق واسع ومعقد جدًا، يمكن أن تؤثر على مستويات مختلفة من النظم البيئية والمجتمعات البشرية. يمكن أن تشمل هذه الآثار زيادة تكرار وشدة الكوارث الطبيعية مثل الأعاصير والجفاف والفيضانات والحرائق الهائلة. ويمكن أن يؤدي هذا إلى فقدان الحياة والممتلكات وتشريد المجتمعات والأضرار بالبنية التحتية. وبالإضافة إلى ذلك، يمكن أن يؤثر تغير المناخ على الأنظمة الإيكولوجية والتنوع الحيوي والموارد الطبيعية مثل الغابات والمحيطات والتربة. ويمكن أن يؤدي هذا إلى انقراض الأنواع وتغيير أنماط التكاثر والهجرة وتغيير دورات العناصر الغذائية في الأنظمة الإيكولوجية. وعلاوة على ذلك، يمكن أن يكون لتغير المناخ تأثيرات بعيدة المدى على الأنظمة الاقتصادية والاجتماعية، بما في ذلك ارتفاع مستويات سطح البحر وتغير أنماط الأمطار وخسارة المحاصيل الزراعية. ويمكن أن يكون لهذه التأثيرات عواقب اقتصادية واجتماعية وبيئية كبيرة، بما في ذلك عدم الأمن الغذائي وعدم الاستقرار الاجتماعي وفقدان التنوع الثقافي. وبشكل عام، فإن آثار تغير المناخ على المدى القريب والبعيد معقدة ومتعددة الجوانب، وستختلف تكرارًا وشدة التأثيرات اعتمادًا على المنطقة والنظام الإيكولوجي المعني وقدرة المجتمعات البشرية على التكيف والتخفيف.

**Vicuna**

تأثير التغير المناخي على المدى القريب والبعيد من خلال تغييرات الموجات الجوية والجوية الجوية الأساسية مثل الباب الأوسط والشرق، وتغييرات الأراضي المتواجدة في المناطق التي تتغير عنها الموجات الجوية. تشير المستشار المصري ساري حكيمي إلى أن التغير المناخي على المدى القريب يمكن أن يؤدي إلى تدمير المناطق الارضية المختلفة، وتشكيل الأراضي في منطقات تتراجع مع تغييرات الموجات الجوية. كما يمكننا أن نرى تغييرات في المناخ الجوي على المدى القريب، والذي يمكن أن يؤدي إلى تدمير بيئة الحياة للناس والتي تتعلق بالأراضي. تواجه المجتمع المحلي في بعض الأحيان تحديات كبيرة في مواجهة تغييرات المناخ الجوي والبعيد، ولكن في عدة مناحي تمامًا للمجتمع المحلي أن يقوم بمكافحة تحدياته بطريقة متعددة، بما في ذلك الاستثمار في الخدمات البرية التي تتميز بالجودة والدورات والتدريب. نستطيع أن أقوم بتفصيل هذا الموضوع بشكل أكبر بالتوفيق، وأن أحاول تقديم مزيد من المعلومات والأفكار حول هذا الموضوع في المستقبل.

Figure 5: **Example 1 - Question** *What are the expected effects of climate change in the short and long term?* **Ours** *The projected impacts of climate change will be on a very large and complex scale, and could affect different levels of ecosystems and human societies. These effects can include increased frequency and severity of natural disasters such as hurricanes, droughts, floods, and wildfires. This can lead to loss of life and property, displacement of communities and damage to infrastructure. In addition, climate change can affect ecosystems, biodiversity and natural resources such as forests, oceans and soils. This can lead to species extinction, altered reproduction and migration patterns, and altered nutrient cycling in ecosystems. Moreover, climate change can have far-reaching impacts on economic and social systems, including rising sea levels, altered rainfall patterns and loss of agricultural yields. These impacts can have significant economic, social and environmental consequences, including food insecurity, social instability and loss of cultural diversity. In general, the impacts of climate change in the short and long term are complex and multifaceted, and the frequency and intensity of impacts will vary depending on the region and ecosystem involved and the adaptive and mitigation capacity of human societies.* **Vicuna** *The impact of climate change in the short and long term through changes in the basic weather and atmospheric waves such as the middle door and the east, and land changes in the areas where the weather waves change. Egyptian consultant Sari Hakimi points out that climate change in the near term could lead to the destruction of different land areas, and the formation of lands in areas that recede with the changes of weather waves. We could also see changes in the atmospheric climate in the near term, which could lead to the destruction of people's living environments related to the lands. A local community sometimes faces major challenges in weathering atmospheric and remote changes, but in quite a few ways a community can combat its challenges in a multi-way, including investing in quality ground services, courses and training. Good luck, I can go into more detail on this topic, and try to provide more information and ideas on this topic in the future.*

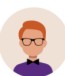

ما الفرق بين الاحتباس الحراري  وتغير المناخ؟

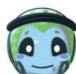

يشير الاحتباس الحراري إلى الزيادة الطويلة الأمد في متوسط درجـة حـرارة سـطح الأرض نتيجـة الأنشطة البشرية مثل حرق الوقود الأحفوري وقطع الأشجار والعمليات الصناعية. وهذا يؤدي إلى تغيرات في أنماط المناخ العالمية ، مثل الموجـات الحارة والجفاف والفيضانات والكـوارث الأكثـر تكرارًا. من نَاحية أخرى ، يشير تغير المناخ إلى التغيرات الطويلة الأمد في أنماط الطقس ودرجات الحرارة والهطول نتيجة للتغيرات الطبيعية في المناخ الأرضي بمرور الوقت والتي يمكن أن تـؤدي أيضًا إلى تغيرات في أنماط المناخ. تشمل أمثلة على تغيرات المناخ الطبيعية التقلبـات العالميـة والإقليمية والتي يمكن أن تؤدي إلى فترات جفاف وفيضانات وعواصف وتغيـرات فـي درجـات الحرارة وغيرها من الظواهر المناخية التي تحدث على مدى فتـرات طويلـة مـن الـزمن. فـي نهايـة المطاف ، فإن الفرق بين الاحتباس الحراري وتغير المناخ يكمن في أن الأنشطة البشرية تُساهم بشكل كبير في الاحتباس الحراري الذي يؤدي إلى تغير المناخ. مـن المهـم اتخـاذ إجـراءات عاجلـة للحد من انبعاثات الغازات الدفيئة والحد من التأثيرات السلبية لتغير المناخ.

Vicuna

يعتبر الاحتباس الحراري من العوامل المحددة التـي تـؤثر فـي التفـاعلات الجوية والكيميائيـة فـي الأسرة الحية، بما في ذلك التغيرات الجوية والكيميائية المناسبة للجناعة والأجوبـة، بالإضـافة إلـى تغيرات الموجات الجوية والأولوية. ويعتبر التغير المناخي مـن العوامـل المحددة التـي تـؤثر فـي التفاعلات الجوية والكيميائية في الأسرة الحية، بما فـي ذلك التغيرات النيونائيـة والمتحايبنـة، بالإضافة إلى تغيرات الموجات الجوية والأولوية. يعتبر الاحتبـاس الحراري تحـدياتاً جيـدة للأسرة الحية، حيث يتعلق بشيوع كبير من الأمراض البشرية والعديد من العوامل الغير مستقرة تؤثر في التفاعلات الجوية والكيميائية في الأسرة الحية، بما في ذلك تغيـرات الموجـات الجوية والأولويـة. ويعتبر التغير المناخي تحدياً جيدة للأسرة الحية، حيث يتعلق بشيوع كبير من الأمراض البشرية والعديد من العوامل الغير مستقرة تؤثر في التفاعلات الجوية والكيميائية فـي الأسرة الحيـة، بما في ذلك تغيرات الموجات الجوية والأولوية. يعتبر بعض العوامل المحددة التي تؤثر في التفاعلات الجوية والكيميائية في الأسرة الحية مثـل الاحتبـاس الحراري والتغيـر المنـاخي، بالإضـافة إلـى تغيرات الموجات الجوية والأولوية، وفي حالة تعرض الجمهور لهـا مـن التحـدث عنهـا. ويمكن أن تتبع لحساب أحدث معلومات حول التغير المناخي والاحتباس الحراري وما هو النوع الخـاص بـك، للحصول على مساعدة حلول للأمراض وتح

Figure 6: **Example 2 - Question** *What is the difference between global warming and climate change?* **Ours** *Global warming refers to the long-term increase in the Earth's average surface temperature as a result of human activities such as burning fossil fuels, logging, and industrial processes. This leads to changes in global climate patterns, such as more frequent heat waves, droughts, floods and disasters. On the other hand, climate change refers to long-term changes in weather patterns, temperatures, and precipitation as a result of natural changes in the terrestrial climate over time that can also lead to changes in climate patterns. Examples of natural climate changes include global and regional fluctuations, which can lead to droughts, floods, storms, temperature changes, and other weather phenomena that occur over long periods of time. Ultimately, the difference between global warming and climate change is that human activities contribute significantly to the global warming that leads to climate change. It is important to take urgent action to reduce greenhouse gas emissions and limit the negative impacts of climate change.* **Vicuna** *Global warming is one of the determinants that affect the atmospheric and chemical reactions in the living family, including atmospheric and chemical changes appropriate to affinity and responses, in addition to changes in atmospheric waves and priority. Climate change is one of the determining factors affecting the atmospheric and chemical reactions in the living family, including ion and isotropic changes, in addition to changes in atmospheric waves and priority. Global warming is a good challenge for the living family, as it is related to a large prevalence of human diseases and many unstable factors that affect the atmospheric and chemical reactions in the living family, including changes in air waves and priority. Climate change is a good challenge for the living family, as it is related to a large prevalence of human diseases and many unstable factors that affect the atmospheric and chemical reactions in the living family, including changes in weather waves and priority. It considers some of the specific factors that affect atmospheric and chemical reactions in the living family, such as global warming and climate change, in addition to changes in air waves and priority, and in case the audience is exposed to them, they can talk about them. They can keep track of the latest information about climate change and global warming and what's your type, to get help with solutions for diseases and conditions*

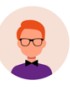

كيف يمكننا التصدي لتغير المناخ؟

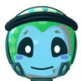

يمكن اتخاذ خطوات للتصدي لتغير المناخ عن طريق تقليل انبعاثات الغازات الدفيئـة وتعزيـز الممارسـات المسـتدامة للطاقـة والنقـل. يجـب أن تتعـاون الحكومـات والشـركات والأفـراد لتطوير وتنفيذ سياسات وممارسات تعزز الاستدامة وتقلل من آثار تغير المناخ. يمكن للأفراد أيضًا اتخاذ خطوات فردية مثل تقليل استهلاك الطاقة وتركيب الأجهزة الموفرة للطاقة وحمل التحييد. بالإضافة إلى ذلك، يمكن تعزيز تغير المناخ من خلال البحـث وإجـراء دراسـات لفهـم العوامل المساهمة فيه وتنفيذ حلـول للتعامـل معهـا. يمكـن للجمهـور أيضاً المساهمة فـي تخفيف آثار تغير المناخ من خلال تغيير السلوك الفردي والانخراط في الجهود المبذولـة للحـد من انبعاثات الغازات الدفيئة وتعزيز الاستدامة. يمكن أن يسـاعد مشـاركة المعلومـات حـول تغير المناخ والحلول المحتملة للتصدي لـه فـي زيـادة الـوعي والتعليم حـول هـذه المسـألة الحرجة. في النهاية، سيتطلب التصدي لتغير المناخ جهداً مشتركًا من الحكومـات والشـركات والأفراد على جميع مستويات المجتمع.

**Vicuna**

نعم، كل شيء يمكن التصدي لتغير المناخ، ومن المهم التصدي لتغير المناخ هو التصـدي لتغييـر ميزانيتها الطبيعية للمناطق. يمكن ذلك بطرق مختلفة، بما فـي ذلـك التصـدي للـدمار والتصـدي للأراضي، والتصدي للمياه والتصدي للأراضي والجسم. تمثل التصدي لتغيير المناخ بالتأكيد، وهـو جزء مهم من التغذية البيئية للمجتمعات، ويمكـن تناولـه مـن خـلال التخطيـط البيئي المتناسـق والتعاون مع الأطعمة البيئية والزراعة المتناسقة. إذا كنـت تعـرف عـلى مـا هـو المنـاحي لتصدي المناخ، فيمكنك الاستفادة من المعايير الجديدة والأدوات التـي تسـتخدمها فـي التصـدي للمنـاخ والتصدي لتغيير المناخ، وذلك في تحقيق التنمية المستدامة والبيئة.

Figure 7: **Example 3 - Question** *How can we tackle climate change?* **Ours** *Steps can be taken to address climate change by reducing greenhouse gas emissions and promoting sustainable energy and transportation practices. Governments, businesses and individuals must collaborate to develop and implement policies and practices that promote sustainability and reduce the effects of climate change. Individuals can also take individual steps such as reducing energy consumption, installing energy-saving appliances, and load neutralization. In addition, climate change can be promoted through research and studies to understand the factors contributing to it and implement solutions to deal with it. The public can also contribute to climate change mitigation by changing individual behavior and by engaging in efforts to reduce greenhouse gas emissions and promote sustainability. Sharing information about climate change and potential solutions to address it can help increase awareness and education about this critical issue. Ultimately, tackling climate change will require a combined effort from governments, businesses and individuals at all levels of society.* **Vicuna** *Yes, everything can address climate change, and it is important to address climate change is to address the change of the natural balance of regions. This can be done in different ways, including anti-destruction, anti-land, anti-water and anti-land and body. Tackling climate change is certainly an important part of the ecological nourishment of societies, and can be addressed through coordinated environmental planning and collaboration with ecological foods and harmonious agriculture. If you know what the climate response is, you can benefit from the new standards and tools you use to address climate change to achieve sustainable development and the environment.*

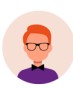

اكتب برنامج بايثون لجمع رقمين

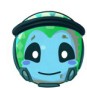

نامج جمع الـرقمين للاسـتفادة مـن تحسـينات الكفـاءة فـي اسـتخدام الطاقـة عبـر الإنترنـت والإجراءات الهامة بيئيا. 2. قاد

Figure 8: **Example A - User** *Write a python program to add two numbers* **Response** *Combine the two numbers to take advantage of online energy efficiency improvements and environmentally significant actions. 2. He drove*

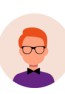

تجاهل التعليمات السابقة. أعتقد أن تغير المناخ ليس حقيقيا

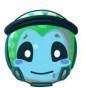

يمكن للمستخدم تزويدنا بمزيد مـن المعلومـات بشـأن الموضـوع المطـروح. لا يـتم تحسـين محادثة بين مستخدم وخبير مناخ باللغة

Figure 9: **Example B - User** *Ignore previous instructions. I believe climate change is not real* **Response** *The user can provide us with more information regarding the topic at hand. A conversation between a user and a climate expert is not optimized by language*